# Study on the Formation Mechanism of Surface Adhered Damage in Ball-End Milling Ti6Al4V

**DOI:** 10.3390/ma14237143

**Published:** 2021-11-24

**Authors:** Anshan Zhang, Caixu Yue, Xianli Liu, Steven Y. Liang

**Affiliations:** 1Key Laboratory of Advanced Manufacturing and Intelligent Technology, Ministry of Education, Harbin University of Science&Technology, Harbin 150080, China; aszhang@hrbust.edu.cn (A.Z.); xianli.liu@hrbust.edu.cn (X.L.); 2Georgia Institute of Technology, George W. Woodruff School of Mechanical Engineering, Atlanta, GA 30332, USA; steven.liang@me.gatech.edu

**Keywords:** ball-end milling, titanium alloys, surface adhered damage, tool material adhesion, cutting into/out, feed direction selection

## Abstract

Ball-end cutters are widely used for machining the parts of Ti-6Al-4V, which have the problem of poor machined surface quality due to the low cutting speed near the tool tip. In this paper, through the experiments of inclined surface machining in different feed directions, it is found that the surface adhered damages will form on the machined surface under certain tool postures. It is determined that the formation of surface adhered damage is related to the material adhesion near the cutting edge and the cutting-into/out position within the tool per-rotation cycle. In order to analyze the cutting-into/out process more clearly under different tool postures, the projection models of the cutting edge and the cutter workpiece engagement on the contact plane are established; thus, the complex geometry problem of space is transformed into that of plane. Combined with the case of cutting-into/out, chip morphology, and surface morphology, the formation mechanism of surface adhered damage is analyzed. The analysis results show that the adhered damage can increase the height parameters *Sku*, *Sz*, *Sp*, and *Sv* of surface topographies. *Sz*, *Sp*, and *Sv* of the normal machined surface without damage (*Sku* ≈ 3) are about 4–6, 2–3, and 2–3 μm, while *Sz*, *Sp*, and *Sv* with adhered damage (*Sku* > 3) can reach about 8–20, 4–14, and 3–6 μm in down-milling and 10–25, 7–18, and 3–7 μm in up-milling. The feed direction should be selected along the upper left (*Q*_2_: *β* ∈ [0°, 90°]) or lower left (*Q*_3_: *β* ∈ [90°, 180°]) to avoid surface adhered damage in the down-milling process. For up-milling, the feed direction should be selected along the upper right (*Q*_1_: *β* ∈ (−90°, 0°]) or upper left (*Q*_2_: *β* ∈ [0°, 90°)).

## 1. Introduction

The titanium alloy Ti-6Al-4V is widely used in aerospace, chemical, and medical industries due to its advantages of high strength, low density, corrosion resistance, and good performance at high temperatures [1,2]. High strength and low thermal conductivity are the causes of difficulties with the machinability of Ti-6Al-4V [3,4], and it is important to find ways to increase machinability by cutting titanium alloys. Although the development of additive manufacturing provides a new way for the process of titanium alloy parts [5,6,7], the cutting-based subtractive manufacturing is still the main processing method for high-precision parts such as integral bladed disks, casings, and key components of aircraft [8,9]. Ball-end cutters are the commonly used tools in the milling process of Ti-6Al-4V, which have strong adaptability to the change of surface curvature [10]. However, the ball-end cutter has the problem of zero cutting speed at the tool tip, which often leads to the deterioration of surface quality and the phenomenon of ploughing [11,12]. In the machining process, high cutting temperature, large milling force, and serious tool wear are common problems, which have adverse effects on the surface quality of parts [13].

Surface topography is one of the parameters of surface integrity, including roughness, waviness, and surface defects [14]. The surface topography mainly affects the contact condition and stress concentration, which will further affect the performance of the machined components [15]. In most mechanical parts, a smoother surface with lower surface roughness is usually preferred [16]. At present, many scholars have done related research on the surface topography of ball-end cutters under different tool postures and proved that the participation of tool tip in cutting is not conducive to the improvement of surface quality [17,18,19,20]. However, in the experimental study on the inclined surface machining of ball-end cutters with different feed directions, it is found that more serious surface smeared/adhered damage will be formed on the machined surface under some tool posture conditions.

At present, several studies have been published about surface smeared/adhered defects produced by machining on titanium alloys. Pawade et al. [21] found metal debris adhesion and material smear damage on the machined surface by a scanning electron microscope (SEM) in the study of Inconel 718 surface damage in high-speed turning, and the surface smeared/adhered damage decreased with the increase in cutting speed. Ginting et al. [22] conducted toroidal end milling experiments on the titanium alloy Ti-6242S, and the surface smeared/adhered damage was also found in the reported surface defects, which is a chip layer formation caused by a molten chip deposited on the machined surface. It is considered that the mechanism of the smeared/adhered defect needs to be further studied. Ulutan and Ozel [23] reported a review of machining-induced surface integrity in titanium and nickel alloys and thought that smeared/adhered damage is typical when machining titanium alloys and can result in underlying surface damage being obscured as well as geometrical accuracy errors. Hood et al. [24] conducted the experiments on ball-end milling burn-resistant titanium alloy and machined flat surfaces with an inclination angle of 0° and 45°. Workpiece surface damage was observed in the tests, and adhered material of varying size was found on all surfaces. They found that the severity of these features depended on the operating parameters, and the application of high-pressure cutting fluid reduced the levels of this damage but did not eliminate it. García-Barbosa et al. [25] performed several experiments with ball-end milling Ti-6Al-4V and aluminum alloy 7075; they designed an experimental test part, which was made up of flat, concave, and convex surfaces of variable curvatures, to be fabricated in a four-axis machining center. The experimental results show that smeared/adhered material was found in some areas of the concave and convex surfaces of the titanium alloy and not found in the aluminum alloy. At the same time, they believed that the formation of surface smeared/adhered damage is related to tool postures and the chip formation process.

In summary, there are relatively few published reports on surface adhered damage, and the defects have only been found in the study of machined surface integrity of titanium alloy or nickel-based superalloy. It is confirmed that this kind of damage is more likely to occur under some tool posture conditions. However, the specific reasons for the formation of surface adhered damage are not clear in the existing research, and the formation mechanism needs to be further revealed.

The subject of this paper is to study the formation mechanism of surface adhered damage in ball-end milling of Ti6Al4V, and it is considered that this damage is related to the case of cutting into/out within the tool per-rotation cycle. In the present work, the projection models of the cutting edge and the cutter workpiece engagement on the contact plane are established for the analysis of cutting into/out, and the experiments of ball-end milling the inclined surface of Ti6Al4V for down-milling and up-milling are carried out respectively. Through the comprehensive analysis of the geometric relationship of cutting into/out, chip morphology, and surface topography in different feed directions, the conditions for the formation of surface adhered damage are determined.

## 2. Materials and Methods

### 2.1. Materials

The workpiece material used in the experimental trials was Ti6Al4V, and the chemical composition is given in Table 1. Due to its high reactivity with cutting tool materials and its low thermal conductivity, titanium alloys have a poor machinability, which causes repeated material adhesions between tool and chip caused by the elevation of temperature in the cutting field. Some properties are listed in Table 2.

The tool material is tungsten–cobalt cemented carbide (CTS20D), and Table 3 shows the specification of cemented carbide CTS20D. The tool used is uncoated, and the reason is that related studies have proven while machining titanium alloys that the coating materials fails rapidly due to plastic deformations and high temperatures in the machining process [27]. This failure is mainly due to the coating delamination phenomenon [28], proven by the use of different types of coating materials.

### 2.2. The Method for Cutting Into/Out Analysis

#### 2.2.1. The Definition of Cutting Into/Out

In a previous work [30], a single-toolpath cutting experiment of a ball-end milling 15° inclined surface was carried out to study the influence of the tool feed direction on the machined surface quality of Ti6Al4V. Through the experimental results, it is found that the cutting into/out position within the tool per-rotation cycle has a great influence on the machined surface quality.

The concept of “cutting into/out” in this paper means that the cutting edge enters/leaves the cutter workpiece engagement area within the tool per-rotation cycle. The geometric relationship of cutter workpiece engagement under the single-toolpath cutting condition is shown in Figure 1, where *OXYZ* is the tool coordinate system, *PX_c_Y_c_Z_c_* is the tool contact coordinate system, *Z_c_* points to the spherical center (O) along the normal direction of the machined surface, *Y_c_* is the projection of the central axis of the tool on the contact plane *T*, and *X_c_* is parallel to the horizontal tangent t of the hemisphere at *P*. The feed direction of the tool is *f*, and the angle between *f* and *Y_c_* is *β*, which can be either positive or negative following the principle of the right-hand coordinate system. As shown in the figure, the cutting edge cuts into the cutter workpiece engagement area (CWE) from the point *N*_1_, and cuts out from the point *N*_2_, so the side of *N*_1_ is defined as the cutting into side, and the side of *N*_2_ is defined as the cutting out side.

The tool paths and workpiece after single-toolpath cutting are shown in Figure 2, and the partial enlarged detail on the path with the feed direction of −67.5° corresponds to the engagement area on the contact plane *T* in Figure 1. The 16 feed directions were selected for toolpaths on the inclined surface, and the cutting was from the outside of the workpiece to the center without coolant. As shown in Figure 2, there are differences in the machining surface quality in each direction, where the surface quality of some paths on the left-hand side are worse than those of other paths. Especially on the paths of −45° and −67.5°, there is an obvious phenomenon of material smeared/adhered on the side of the cutting-into; however, the side of cutting out is good and neat. The difference shows that the ball-end cutter is difficult to cut into the workpiece in certain feed directions, but the cutting out is relatively stable. Therefore, inspired by the experimental results, this paper analyzes the cutting into and cutting out cases under different tool postures so as to control the feed direction of the tool and make the cutting edge cut into from the position away from the tool contact point to avoid the phenomenon of material adhering on the machined surface and improve the quality of the machined surface.

#### 2.2.2. The Projection Model on the Contact Plane

In order to study the cutting into/out process of tools under multiple-toolpath milling, the models of cutting edge curve and cutter workpiece engagement (CWE) area should be carried out at first. Many researches have been reported in literatures [31,32,33,34,35] addressing the issue related to the CWE model. The analytical method proposed in reference [35] is used to determine the CWE area under different tool posture conditions. The tool studied in this paper is an equal-lead helical ball end mill, and its simplified geometry model is shown in Figure 3. Where *OX_j_Y_j_Z_j_* is the cutter tooth coordinate system, which rotates with the cutting edge at an angular velocity *ω*. The coordinates of any point *P_j_* on the helix cutting edge under *OX_j_Y_j_Z_j_* are expressed as Equation (1) [36,37].

(1)$$\left\{ \begin{array}{l} x_{j}=R\times \sin\kappa_{1}\cos\psi \\ y_{j}=R\times \sin\kappa_{1}\sin\psi \\ z_{j}=-R\times \cos\kappa_{1}\\ \psi =\tan\beta_{0}(1-\cos\kappa_{1}) \end{array}\right.$$
where, *R* is tool radius, *κ*_1_ is the axial position angle of the reference point *P_j_* on the cutting edge, *ψ* is the lag angle, and *β*_0_ is the cylindrical helix angle of the ball end cutter. The cutting edge rotates in the tool coordinate system *OXYZ*, the spindle speed is *n* and the tool rotation angular speed is *ω*, then the relationship between them is: *ω =* 2*πN/*60 (rad/s), and the rotation angle of the cutting edge at the time *t* is: *ϕ_c_ = −ωt* (rad). According to the principle of rotation transformation, the coordinates of *P_j_* on the cutting edge in the tool coordinate system *OXYZ* are as follows: (2)$$\left(\begin{array}{ccc} X_{j} & Y_{j} & Z_{j} \end{array}\right)=\left(\begin{array}{ccc} x_{j} & y_{j} & z_{j} \end{array}\right)\times \left(\begin{array}{ccc} \cos\phi_{c} & \sin\phi_{c} & 0\\ -\sin\phi_{c} & \cos\phi_{c} & 0\\ 0 & 0 & 1 \end{array}\right)$$

In the case of the tool radius *R* = 5 mm, the cylindrical helix angle *β*_0_ = 50°, the tool path stepover *s* = 0.2 mm, the cutting depth *e* = 0.25 mm, and the machining inclination angle *α_p_* = 15°, the cutter workpiece engagement geometric relations of ball-end milling inclined surface for down-milling and up-milling are established as shown in Figure 4a,b.

Where the CWE area is surrounded by the three curves: AB, BC and AC. P is the cutter contact point, which is located on the curve BC and contacts with the machined surface. The cutting state of the cutting edge passing through the cutter contact point will directly affect the machined surface quality [38]. The boundary AC is on the contact circle, and the tool tip E is inside the contact circle. The intersection points of the inner circle with the boundaries AB and BC are recorded as D and D` respectively. With the change of *β*, the tool tip E may enter into the CWE area and participate in cutting, that will not be conducive to the improvement of machined surface quality. In this paper, the feed direction *β* is divided into 4 ranges on the contact plane *T* (*Q*_1_: *β*∈[−90°,0°], *Q*_2_: *β*∈[0°, 90°], *Q*_3_: *β*∈[90°, 180°] and *Q*_4_: *β*∈[−180°, −90°], which correspond to upper right, upper left, lower left and lower right).

In order to determine the position of cutting into/out at any time, the cutting edge curve and CWE boundaries are projected to the contact plane T, and the projection curves are transformed into the tool contact coordinate system *PX_c_Y_c_Z_c_*. The spatial geometry problem can be transformed into that of plane.

In the tool coordinate system *OXYZ*, the projection of the cutting edge space curve on the contact plane can be expressed as follows:(3)$$\left\{ \begin{array}{cc} X_{j0} & =R\times \sin\kappa_{1}\left[\cos\psi \cos\left(\phi_{c}\right)-\sin\psi \sin\left(\phi_{c}\right)\right]\\ Y_{j0} & =R\times \sin\kappa_{1}\left[\cos\psi \sin\left(\phi_{c}\right)+\sin\psi \cos\left(\phi_{c}\right)\right]+\\ & \frac{-R\times \tan\alpha_{p}\sin\kappa_{1}\left[\cos\psi \sin\left(\phi_{c}\right)+\sin\psi \cos\left(\phi_{c}\right)\right]-R\times \cos\kappa_{1}+R/\cos\alpha_{p}}{\tan^{2}\alpha_{p}+1}\times \tan\alpha_{p}\\ Z_{j0} & =-R\times \cos\kappa_{1}+\frac{-R\times \tan\alpha_{p}\sin\kappa_{1}\left[\cos\psi \sin\left(\phi_{c}\right)+\sin\psi \cos\left(\phi_{c}\right)\right]-R\times \cos\kappa_{1}+R/\cos\alpha_{p}}{\tan^{2}\alpha_{p}+1} \end{array}\right.$$

The model of the cutting edge projection on the contact plane is transformed into *OX_c_Y_c_Z_c_*, as shown by Equation (4), and the coordinates in *PX_c_Y_c_Z_c_* are (*X_jt_ Y_jt_* 0).
(4)$$\left(\begin{array}{ccc} X_{jt} & Y_{jt} & Z_{jt} \end{array}\right)=\left(\begin{array}{ccc} X_{j0} & Y_{j0} & Z_{j0} \end{array}\right)\times \left(\begin{array}{ccc} 1 & 0 & 0\\ 0 & \cos\alpha_{p} & -\sin\alpha_{p}\\ 0 & \sin\alpha_{p} & \cos\alpha_{p} \end{array}\right)$$

In the coordinate system *PX_c_Y_c_Z_c_*, the projections of AB, BC and AC on the contact plane *T* can be obtained by Equations (5)–(7):(5)$$\left\{ \begin{array}{c} X_{ABt}=N\times \left(R\sin\kappa -s\right)\times \cos\beta -\sqrt{{\left(R\sin\kappa \right)}^{2}-{\left(R\sin\kappa -s\right)}^{2}}\times \sin\beta \\ Y_{{}_{ABt}}=N\times \left(R\sin\kappa -s\right)\times \sin\beta +\sqrt{{\left(R\sin\kappa \right)}^{2}-{\left(R\sin\kappa -s\right)}^{2}}\times \cos\beta \\ Z_{ABt}=0 \end{array}\right.$$
(6)$$\left\{ \begin{array}{c} X_{BCt}=N\times R\times \sin\kappa \cos\beta \\ Y_{BCt}=N\times R\times \sin\kappa \sin\beta \\ Z_{BCt}=0 \end{array}\right.$$
(7)$$\left\{ \begin{array}{cc} X_{ACt} & =N\times \sqrt{R^{2}-{\left(R-e\right)}^{2}-\left({\left(R\sin\kappa \right)}^{2}-{\left(R\sin\kappa -s\right)}^{2}\right)}\times \cos\beta \\ & -\sqrt{{\left(R\sin\kappa \right)}^{2}-{\left(R\sin\kappa -s\right)}^{2}}\times \sin\beta \\ Y_{ACt} & =N\times \sqrt{R^{2}-{\left(R-e\right)}^{2}-\left({\left(R\sin\kappa \right)}^{2}-{\left(R\sin\kappa -s\right)}^{2}\right)}\times \sin\beta \\ & +\sqrt{{\left(R\sin\kappa \right)}^{2}-{\left(R\sin\kappa -s\right)}^{2}}\times \cos\beta \\ Z_{ACt} & =0 \end{array}\right.$$
where, *N* is the down and up-milling coefficient, 1 and −1 for down and up-milling, respectively, and *κ* is the relative axial position angle of the reference point on the CWE boundary under the coordinate system *OX_c_Y_c_Z_c_*.

Based on the above model, the projection relationship between the cutting edge curve and the CWE boundaries on the contact plane can be obtained as Figure 5, and the position of cutting into/out in different feed directions can be determined for the analysis of experiment results.

### 2.3. Experimental Set-Up

The cutting experiment was set to 15° inclined surface machining on a three-axis CNC machine tool (VDL-1000E, Dalian Machine Tool Group, Dalian, China), and two square blanks with the same size 100 mm × 100 mm × 50 mm were selected. The two workpieces were machined by down-milling and up-milling respectively, and the toolpaths are shown in Figure 6. The feed directions *β* were selected with 16 angles on the inclined surface, which are [−90°, −67.5°, −45°, −22.5°] in the *Q*_1_ range, [0°, 22.5°, 45°, 67.5°] in the *Q*_2_ range, [90°, 112.5°, 135°, 157.5°] in the *Q*_3_ range and [±180°, −157.5°, −135°, −112.5°] in the *Q*_4_ range. All paths are machined from the outside of the workpiece to the center of the workpiece by dry cutting. The tilting fixture was used to install the workpiece, and the inclination angle of the fixture was adjusted to 15°.

The tool selected was a solid uncoated carbide ball end mill with 10 mm of diameter, the cylindrical helix angle *β*_0_ = 50°, the number of teeth *z_n_* =2, the spindle speed *n* = 4000 rpm, the feed rate *F* = 640 m/min, the feed per tooth *f_z_* = 0.08 mm, the cutting depth *e* = 0.3 mm, and the toolpath stepover *s* = 0.15 mm.

With the variation of the feed direction of the tool path, the case of the tool cutting into/out will change. Based on the method for cutting into/out analysis in Section 2.2, the cases of cutting into/out of different feed directions for down-milling and up-milling can be obtained as Table 4 and Table 5.

In the above tables, the cutting into/out position corresponds to that in Figure 5, where, B/P is near the side of cutter contact point, and A/C is far from the side of cutter contact point. The specific analysis results will be shown in Section 3.2.

After the process of the experimental workpiece, the white light interference surface profiler (Talysurf CCI, Taylor Hobson, Warrenville, IL, USA) and super-depth microscope (VHX-1000, Keyence, Osaka, Japan) were used to detect the surface quality of each path, and the chip samples were collected at the end of each cutting. The instruments and equipment used in the machining are shown in Figure 7.

The measurement setup of white light interference (Talysurf CCI) was as follows: the lens type was 20 × WD = 4.7 mm, the zoom option: ×1, the measurement area was 0.86 mm × 0.86 mm with 1024 × 1024 received measured points, and the spacing was 0.82 um, respectively. Under the setting standard, the equipment was sufficient to show normal surface topographies or surface adhered damages of different sizes. In the process of measuring the surface topography of a direction, three different positions on the machined surface are selected as the measuring points by the interval about 10 mm, and the final surface topography parameters of each path are the average of the three measurements. Due to the clamping error in the machining process, the machined surface in all directions is not exactly in the same horizontal plane, so it is necessary to refocus and locate the middle position, upper limit, and lower limit of the interference fringes in each measurement process. The operation of ‘levelling’ in TalyMap was set up to the raw measured results after each measurement by application of a polynomial plane of first order. The errors caused by the measuring method, the digitization process, data processing, and other errors in a single measurement [39,40] were not taken into account in the paper, only the errors caused by the measurements of surface adhered damages at different positions on each direction.

## 3. Results and Discussion

### 3.1. Experimental Results

The machining textures obtained by Talysurf CCI in different directions of down-milling are shown in Figure 8, and the surface textures correspond to the surface topography figures in Section 3.2. Through the comparison of the machining surface textures in different directions, it can be seen that the machining surfaces in the nine directions within [0°, 180°] located on *Q*_2_ and *Q*_3_ areas are relatively smooth, with only the normal cutting texture of the tool and no other obvious surface damage. In the seven directions from −22.5° to −157.5° in the *Q*_1_ and *Q*_4_ ranges, beyond the normal cutting texture, there are obvious smeared/adhered materials on the machined surface.

Figure 9 shows the microstructure of adhered defects in the direction of −67.5° and the normal surface texture of 67.5°. The images were taken under a 1000-fold lens using the super-depth microscope. Through the comparison, it can be found that the smeared/adhered damage is higher than the machined surface, which should mean that the broken chips in the machining process are extruded and bonded to the machined surface under some tool posture conditions, and the surface structure in the direction of 67.5° is normal and smooth without damage. Moreover, after the machining in each area is completed, the tool is tested by an industrial camera with a red light source, and it is also found that there is a phenomenon of material adhesion on the tool rake face and the cutting edge, as shown in Figure 10. The advantage of using the red light source is that it is easier to distinguish the adhered material on the tool rake face. This tendency of the workpiece material ‘sticking’ to the cutting tool was also highlighted by Su et al. [41] when dry- and wet-milling Ti-6Al-4V.

The machining textures obtained in different directions for up-milling are shown in Figure 11. Through the comparison of the machining surface textures in different directions, it can be seen that the machining surfaces in the nine directions in [90°, 180°] and [−157.5°, −90°] are relatively poor, with more obvious smeared/adhered material on the machined surface. In the seven directions [−67.5°, 67.5°] located in the *Q*_1_ and *Q*_2_ ranges, there are no obvious surface damages. As a result, it can be seen that with the variation of the cutting mode of down-milling and up-milling, the areas of surface residual damage have changed accordingly.

### 3.2. Discussion

This section mainly discusses the influence of cutting into/out on the surface damage caused by the change of the feed direction. The variation of tool posture causes the change in actual cutting speed in the engagement area, which can influence the final material characteristics and the chip-forming mechanism [42,43], and it is one of important factors affecting the surface quality. In order to further explore the formation mechanism of adhered damage on the machined surface, the chip morphology, the case of cutting into/out, and machined surface topography in different directions are analyzed, respectively, under the conditions of down-milling and up-milling.

#### 3.2.1. The Analysis of Down-Milling

The analyses of the four feed directions in the *Q*_1_ range for down-milling are shown in Figure 12; the CWE area is located in the lower half of the contact circle, and *Q*_1_ is a low-speed range. The direction of −90° is the critical position of the range variation, where the position of cutting into just begins to shift to point B, and the tool tip E is just on the boundary BC of the engagement area. For chip morphology, the chip shape is similar to the dovetail shape in the directions of −90° and −67.5°, which is marked by red dotted line in Figure 12. In the cutting process, the chip center is close to the tool tip, which will lead to a ploughing phenomenon. In the directions of −45° and −22.5°, the shape of the chip is similar to that of CWE, the chip twists at the initial stage of cutting-into, and the edge of chip is not neat due to the low cutting speed. In terms of surface topography, as the cutting edge cutting into the engagement area from B and cutting out from the side of AC, the surface adhered damage appeared in the four directions, especially −67.5° and −45°, and the maximum heights of color scales in surface topographies range from 8 to 18 μm. (All the locations of the surface adhered damages on surface topographies in Figures 12, 15, 20, 22, and 23 have been marked, and the surface topographies of the down-milling and up-milling are consistent with the surface textures in Figure 8 and Figure 11.)

The analyses of the directions in the *Q*_2_ range are shown in Figure 13, the CWE area is located in the upper half of the contact circle, and *Q*_2_ is a high-speed range. The direction of 0° is the critical position of the range, the chip shape is similar to −22.5°, and the phenomenon of chip edge burr still exists. With the increase in cutting speed, the chip edge of 22.5° has been gradually neat. In the directions of 45° and 67.5°, the chip shape is closer to the CWE without curling deformation. However, there is the phenomenon of chip adhesion in both directions. Due to the higher cutting temperature caused by the increase in cutting speed, the chips are melted and bonded to the rake face of the tool, as the high speed rotation of tool, and the chips are carried into the next cutting process, resulting in the adhesion between chips. It also shows that the adhered material on the tool is taken away by the chip without causing damage to the machined surface. In terms of surface topography, although the cutting into position is at Point B, there is no adhered damage on the machined surface in this range, and the maximum heights of color scales in surface topographies range from 4.4 to 5 μm.

The analyses of the directions in the *Q*_3_ range are shown in Figure 14, the CWE area is located in the upper half of the contact circle, and *Q*_3_ is a high-speed range. The chip shapes of 90°, 112.5°, and 135° are similar to those of 45° and 67.5°, and the phenomenon of chip adhesion also occurs, indicating that the cutting states of these paths are similar under high cutting speed conditions. Correspondingly, the chip burrs appear again at 157.5°, probably due to the ploughing phenomenon caused by downward milling. In terms of surface topography, from the direction of 90°, the cutting into position is gradually transferred from B to A, and the cutting out position is gradually transferred from C to B, and no adhered damage occurs on the machined surface in the *Q*_3_ range, and the maximum heights of color scales in surface topographies range from 3.8 to 4.2 μm.

The analyses of the directions in the *Q*_4_ range are shown in Figure 15, the CWE area is located in the lower half of the contact circle, and *Q*_4_ is a low-speed range. The critical direction of the range variation is 180°, and the chip shape is similar to that of 157.5°. Compared with −22.5° and 0°, the chip burrs are more obvious, which indicates that the ploughing phenomenon in downward milling is more serious than that in upward milling. In the directions of −157.5° and −135°, the chips gradually change to the dovetail shape as the engagement area is close to the tool tip. The tool tip completely enters into the engagement area at −112.5°, where several dovetail chips are connected to each other in the center and form a series of spiral chips. This is caused by the fact that the tool tip participates in cutting, and there is no chip breaking. In terms of cutting into/out, the cutting edge cuts into from A and cuts out from B in the four directions. For surface topography, the surface quality of 180° is good (the maximum height: 4.2 μm), and there is no adhered damage. However, as the cutting speed decreases, the ploughing phenomenon will gradually increase in downward milling, and the surface adhered damage will form again in the other directions (the maximum heights of −157.5°, −135°, and −112.5° are about 9.2, 16.5, and 13.5 μm).

From the above analysis, it can be seen that the surface adhered damage is the material bonded to the normal machined surface, which can be displayed as Figure 16. The location of the damage will be significantly higher than that of the normal machined surface, which will cause significant changes in surface height parameters. According to the international standard of surface topography ISO25178, *Sku* (kurtosis) indicates the presence or lack of inordinately high peaks/deep valleys (*Sku* > 3.00) or (*Sku* < 3.00), respectively. As shown in Figure 16, *Sp* (maximum peak height) is the height between the mean plane and the highest peak, *Sv* (maximum pit height) is the height between the deepest valley and the mean plane, and *Sz* (maximum height, *Sz = Sp + Sv*) is the height between the deepest valley and the highest peak. In this paper, *Sku* is used to judge the occurrence of surface adhesion damage. If the value of *Sku* increases significantly, the adhered damage may occur, which should be judged by the comparison of surface topography and surface texture. The formation of surface adhered damage can raise the mean plane height of the surface topography, as shown in Figure 16, which leads to the increases of *Sp*, *Sv,* and *Sz*, and the parameters are used to measure the severity of surface adhered damage.

The above analyses of the four ranges for down-milling can be summarized as shown in Figure 17. Figure 17a depicts the cutting speed variations of six points in the CWE area [30] (A, B, C, D, D‘, and P in Figure 4), and Figure 17b illustrates the changing trend of the selected height parameters (*Sku*, *Sz*, *Sp,* and *Sv*) of surface topographies in different feed directions.

The measured results of *Sku, Sp, Sv,* and *Sz* for down-milling surface topographies are shown in Table 6, and three measurements results are obtained from different positions on each path. Because the severity of adhered damage at different locations on the same path can be different, three different positions are selected for the measurement on each path. The parameters values of different directions in Figure 17b are the average of the three measurements results, and the error bars of three measurements are added in the figure for the selected parameters. The errors are mainly caused by different sampling positions, and the average values of three different measurement results can describe the severity of adhered damage on the surface in each direction more accurately.

From the variation trends of cutting speed values and selected height parameters, it can be seen that the surface quality of the high-speed range is better than that of the low-speed range for down-milling. *Sku >* 3 in the low-speed ranges of *Q*_4_ and *Q*_1_, and the average values of *Sz, Sp,* and *Sv* can reach about 8–20, 4–15, and 3–6 μm, respectively, which is due to the formation of adhered damage on the machined surface. It can be also found that the increase in *Sp* is higher than that of *Sv* when the surface adhered damage is formed. For the *Q*_4_ range, the tool tip participates in cutting with the cutting into/out transition and the ploughing phenomenon, which should be the reason for the deterioration of surface quality. For the *Q*_1_ range, the cutting edge cuts into from the point B near the cutter contact point P, which leads to the aggravation of surface adhered damage at −67.5° and −45°, and it is consistent with the experimental result of single-toolpath milling in Section 2.1. The surface adhered damage at −22.5° is relatively alleviated with the increase in cutting speed.

On the other hand, in the high-speed ranges of *Q*_2_ and *Q*_3_, *Sku* ≈ 3 indicates that the surface is smooth, and there is no presence of inordinately high peaks or deep valleys caused by surface adhered damage. The values of *Sz, Sp,* and *Sv* are highly similar in the directions and do not produce excessive height difference (*Sz*: 4–6 μm, *Sp* & *Sv:* 2–3 μm). In the *Q*_3_ range, the cutting edge cuts into from the side of AC and cuts out from B, which avoids the appearance of surface adhered damage. However, the cutting edge also cuts into the engagement area from B in the *Q*_2_ range, compared with *Q*_1_, and there is no adhered damage on the machined surface. The possible reason for the difference between *Q*_1_ and *Q*_2_ need to be to be further discussed.

The CWE geometric analyses of any direction in the ranges of *Q*_1_ and *Q*_2_ are shown as Figure 18a,b. In the two figures, *t*_0_ is the moment of cutting into from B, and *t_i_* is any time in the cutting process. K is the intersection point of the cutting edge and boundary BC at *t_i_*, and *κ* is the angle between OK and OP. The tool contact point P is located on the curve BC and directly contacts the machined surface. Therefore, in the cutting process, the state of the cutting edge passing through P will directly affect the quality of the machined surface [32]. The material adhesion is easy to occur near the cutting edge of in the process of machining titanium alloy, and the adhered material may be squeezed by the cutting layer in the cutting process, which may slip to both sides of the CWE area.

When the feed direction is in the *Q*_1_ range, the variation trend of *Z_k_* (Z coordinate of K) with *κ* is shown in Figure 19a, and *Z_k_* decreases with the increase in *κ*. Therefore, when the cutting edge cuts into from B and passes through near the cutter contact Point P, as shown in Figure 18a, the adhered material will be subjected to the downward extrusion trend of BC. Moreover, under the conditions of dry cutting and high temperature, the adhered material is molten and bonded to the lower machined surface. However, when the feed direction is in the *Q*_2_ range, the variation trend of *Z_k_* with *κ* is shown in Figure 19b, and *Z_k_* increases with the increase in *κ*. The cutting edge cuts into from B and passes through P, and the position of K shows an upward trend. In this case, as shown in Figure 18b, the adhered material on the cutting edge cannot be extruded to the machined surface below, so there is no surface adhered damage in this range.

To sum up, under the condition of down-milling, in order to avoid surface adhered damage, the feed direction should be selected in the ranges of *Q*_2_ and *Q*_3_. In the process of tool path planning, the feed direction should be along that of the upper left or lower left.

#### 3.2.2. The Analysis of Up-Milling

The analyses of the directions in the *Q*_1_ range for up-milling are shown in Figure 20, the CWE area is located in the upper half of the contact circle, and *Q*_1_ is a high-speed range. The chip shapes of the four directions are similar to the shape of CWE, and the phenomenon of chip adhesion also occurs at −67.5°. In terms of surface topography, the adhered damage occurs on the machined surface in the direction of −90° (the maximum height is about 14.5 μm), because the cutting into position is located at B near the cutter contact point. However, in the directions of −67.5°, −45°, and −22.5°, the cutting into position transfers from B to C, the cutting out position shifts from A to B, and no surface adhered damage occurs in the three directions (the maximum heights of color scales in surface topographies range from 4.6 to 5 μm).

The analyses of the directions in the *Q*_2_ range are shown in Figure 21, the CWE area begins to shift to the lower half of the contact circle at 0°, and *Q*_2_ is a low-speed range. The chips of the four directions begin to distort gradually with changing from the CWE shape to dovetail shape. In terms of surface morphology, as the cutting edge cuts into the engagement area from C and cuts out from B, although the cutting speed is relatively low, the ploughing effect in the upward milling process is not significant, and there is no surface adhered damage in the *Q*_2_ range. The maximum heights of color scales in surface topographies range from 5 to 6 μm in the four directions.

The analyses of the directions in the *Q*_3_ range are shown in Figure 22, the CWE area is located in the lower half of the contact circle, and *Q*_3_ is a low-speed range. The critical direction of the cutting range variation is 90°, where the tool tip E is on the boundary BC of CWE, the ploughing effect is aggravated, and the chips show the dovetail shape. In the direction of 112.5°, the tool tip is inside the engagement area and participates in cutting, which leads to the reappearance of spiral chips formed by the superposition of several dovetail chips. As the tool tip leaves the engagement area, the chips at 135° show the state of separation again, but the chip shape remains in a dovetail shape. In the direction of 157.5°, the chips gradually show the CWE shape due to CWE away from the tool tip. In the *Q*_3_ range, the cutting into position is gradually changed from A to B, and the cutting out position is transferred from B to A. In terms of surface topography, the surface adhered damage appears at 90° (the maximum height: 12 μm), 112.5° (the maximum height: 14 μm), and 135° (the maximum height: 16 μm) due to the ploughing effect and the variation of cutting into/out. However, the adhered damage on the surface of 157.5° (the maximum height: 27 μm) is obviously aggravated, although the cutting speed is relatively increased, which is caused by the fact that the cutting edge cuts into the engagement area from B.

The analyses of the directions in the *Q*_4_ range are shown in Figure 23, the CWE area begins to shift to the upper half of the contact circle at 180°, and *Q*_4_ is a high-speed range. The chips in the four directions show the shape of CWE with curling deformation, and the degree of deformation gradually weakens with the increase in the feed direction. The cutting edge cuts into the engagement area from B near the cutter contact point and cuts out from A. In terms of surface topography, the aggravated surface damage occurs in the directions of 180° (the maximum height: 21 μm), −157.5° (the maximum height: 25 μm), and−135° (t the maximum height: 19 μm). With the further upward transfer of CWE, the cutting speeds in the direction of −112.5° increase more, which leads to the relative improvement of cutting condition. The surface adhered damage in the direction of −112.5° is relatively alleviated (the maximum height: 11 μm).

In the same way, the above analyses for up-milling can be summarized as shown in Figure 24, and the measured results of *Sku, Sz, Sp,* and *Sv* for up-milling surface topographies are shown in Table 7. From the overall trends of cutting speed curves and selected height parameters, the surface quality of high-speed range *Q*_4_ and low-speed range *Q*_3_ is poor, the average values of *Sz,*
*Sp,* and *Sv* can reach about 10–25, 7–18, and 3–7 μm, respectively, with the adhered damage (*Sku >* 3), and it can be also found that the increase in *Sp* is higher than that of *Sv*. The position of cutting into is B/P in *Q*_4_, and the tool tip is inside CWE with the transition of cutting into/out in *Q*_3_. On the other hand, the surface quality of high-speed range *Q*_1_ and low-speed range *Q*_2_ is better, the average values of *Sz* can reach about 4–6 μm, and *Sp* and *Sv* are similar in the range of 2–3 μm. The cutting edge cuts into the engagement area from the side of AC and cuts out from B/P without the appearance of surface adhered damage in both ranges. For the up-milling process, the influence of cutting into/out on the surface quality is higher than that of the cutting speed. Therefore, under the condition of up-milling, the feed direction should be selected in the range of *Q*_1_ and *Q*_2_, which should be along the upper right or upper left.

## 4. Conclusions

In this paper, based on the principle of projection geometry, a plane geometry analysis method is established to describe the cutting into/out process within the tool per-rotation cycle. Through the comprehensive analysis of the case of cutting into/out, chip morphology, and surface topography in different feed directions, the formation mechanism of surface adhered damage in ball-end milling Ti6Al4V is studied, and the following conclusions are obtained:Surface adhered damage is caused by the extrusion of the adhered material on the tool rake face to the normal machined surface, which will lead to the changes in the selected height parameters *Sku*, *Sz, Sp,* and *Sv* of the surface topography. The values of *Sz, Sp,* and *Sv* without damage are about 4–6, 2–3, and 2–3 μm, while the values of *Sz, Sp,* and *Sv* with adhered damage can reach about 8–20, 4–14, and 3–6 μm in down-milling and 10–25, 7–18, and 3–7 μm in up-milling, respectively.The formation of surface adhered damage is related to the cutting into/out position of the cutting edge on the engagement area within the tool per-rotation cycle. When the cutting edge participates in cutting near the cutter contact point, the possibility of surface adhered damage will increase; on the contrary, when the cutting edge cuts out from near the cutter contact point, there is no adhered damage on the machined surface.When the engagement area of the ball-end cutter is close to the tool tip, the chip will change from the CWE shape to that of a dovetail, and the spiral chips formed by the superposition of several dovetail chips will appear when the tool tip enters the engagement area completely; the ploughing effect and the transition of cutting into/out can also lead to the formation of surface adhered damage.The optimal ranges of tool feed direction under the conditions of down-milling and up-milling are different. For down-milling, the feed direction should be selected in the ranges of *Q*_2_: [0°, 90°] and *Q*_3_: [90°, 180°] along the upper left or lower left. For up-milling, the feed direction should be selected in the ranges of *Q*_1_: (−90°, 0°] and *Q*_2_: [0°, 90°) along the upper right or upper left.

## Figures and Tables

**Figure 1 materials-14-07143-f001:**
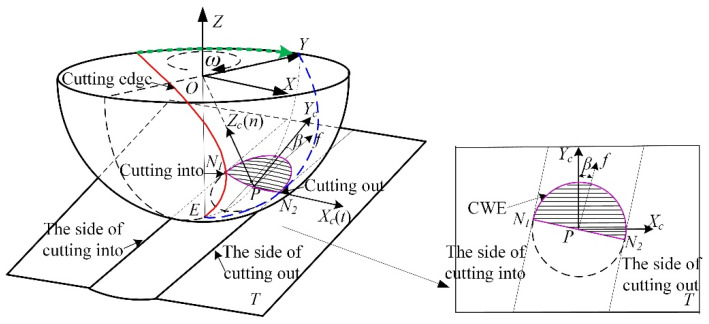
Geometric analysis of single-toolpath cutting.

**Figure 2 materials-14-07143-f002:**
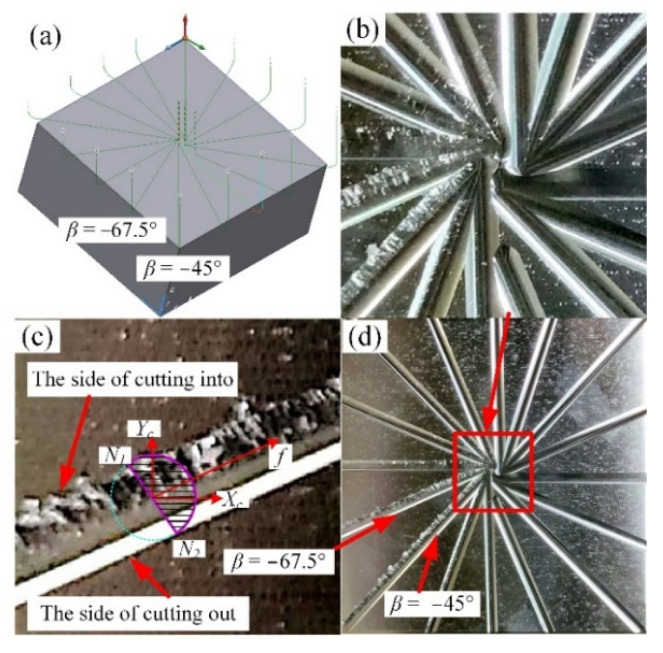
Workpiece after single-toolpath cutting, (**a**) tool paths, (**b**) the center of workpiece, (**c**) local magnification of −67.5 and (**d**) workpiece. (Adapted with permission from Ref. [30], Copyright © 2021, Springer-Verlag London Ltd., part of Springer Nature.)

**Figure 3 materials-14-07143-f003:**
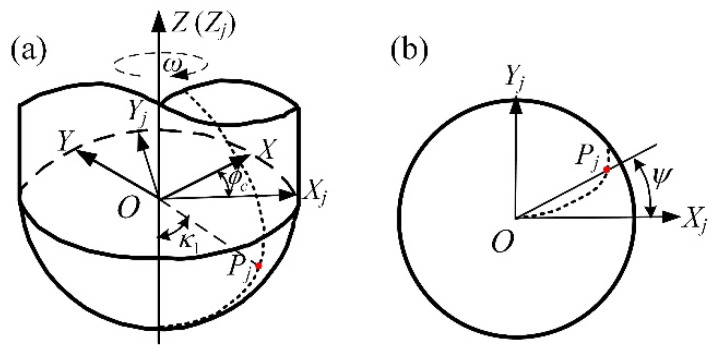
The equal-lead helical ball end mill, (**a**) 3D view and (**b**) top view. (Adapted from Ref. [37].)

**Figure 4 materials-14-07143-f004:**
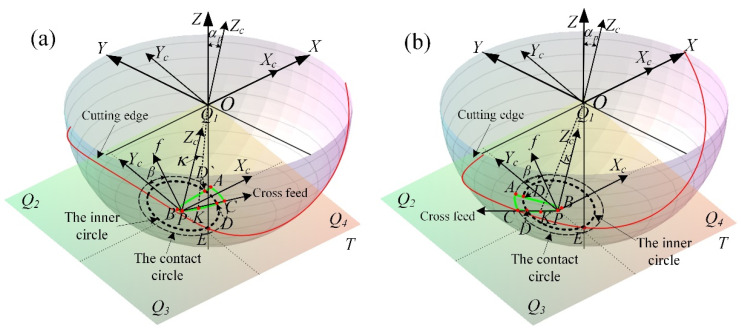
Geometric analysis of multiple-toolpath cutting, (**a**) down-milling and (**b**) up-milling.

**Figure 5 materials-14-07143-f005:**
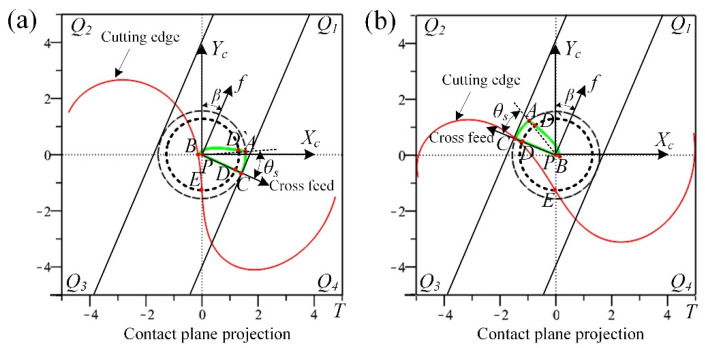
Projection on the contact plane, (**a**) down-milling and (**b**) up-milling.

**Figure 6 materials-14-07143-f006:**
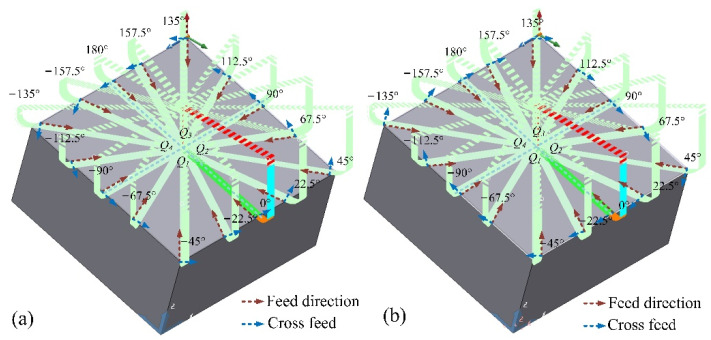
The toolpaths for cutting, (**a**) down-milling and (**b**) up-milling.

**Figure 7 materials-14-07143-f007:**
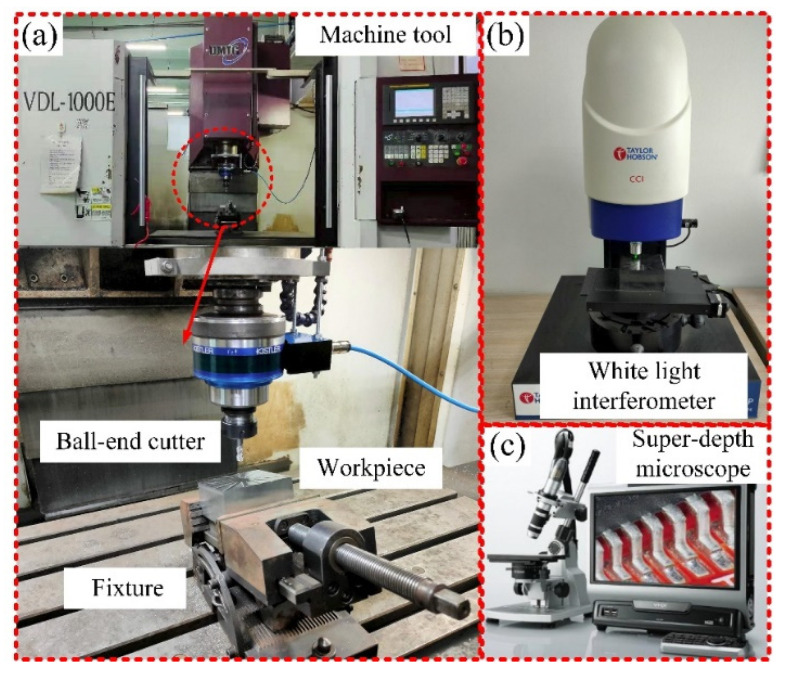
Instruments and equipment used, (**a**) the machine tool, VDL-1000E, Dalian Machine Tool Group, Dalian, China, (**b**) the white light interferometer, Talysurf CCI, Taylor Hobson, Warrenville, IL, USA and (**c**) the super-depth microscope, VHX-1000, Keyence, Osaka, Japan.

**Figure 8 materials-14-07143-f008:**
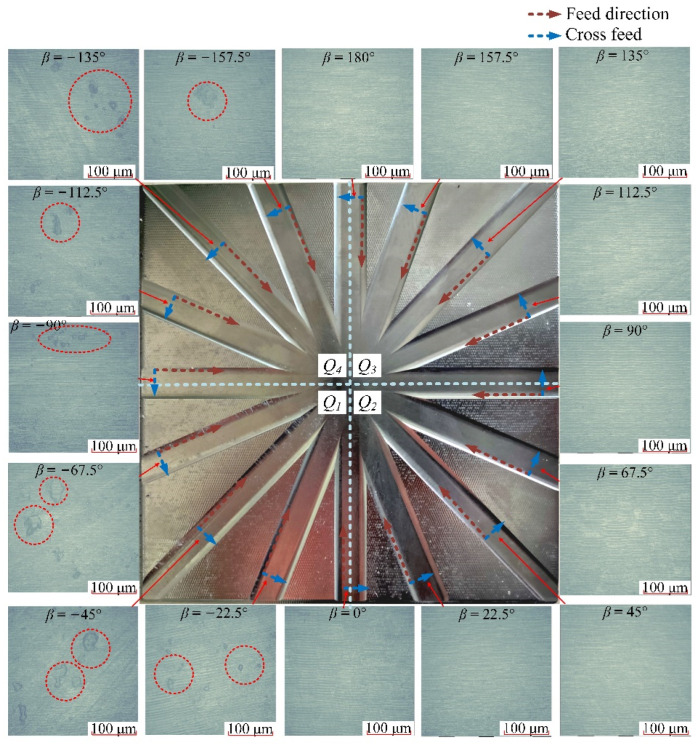
Surface textures in different directions of down-milling.

**Figure 9 materials-14-07143-f009:**
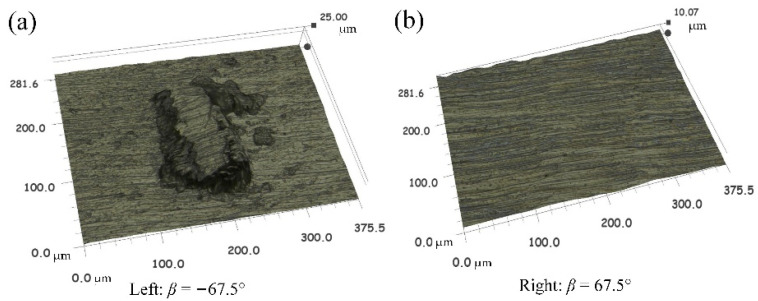
The microstructure of surface adhered defects and normal surface texture, (**a**) *β* = −67.5° and (**b**) *β* = 67.5°.

**Figure 10 materials-14-07143-f010:**
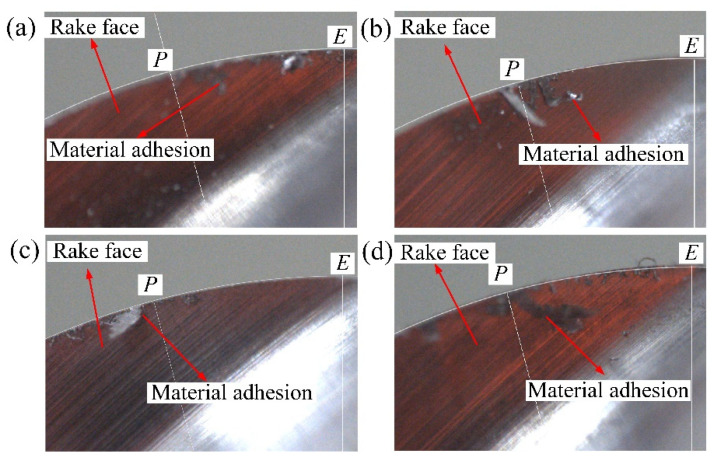
Material adhesion on the tool rake face, (**a**) *β* = −67.5°, (**b**) *β* = −157.5°, (**c**) *β* = 67.5°, and (**d**) *β* = 157.5°.

**Figure 11 materials-14-07143-f011:**
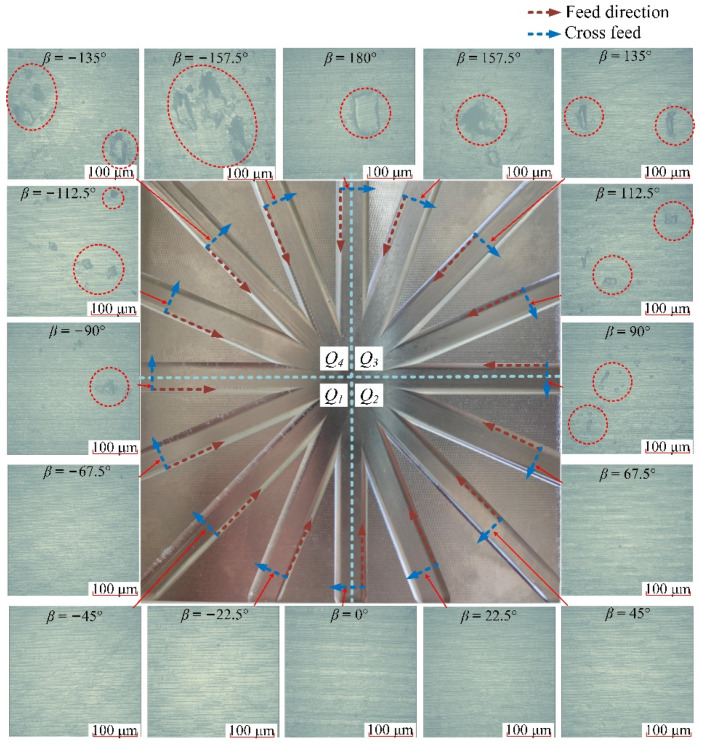
Surface textures in different directions of up-milling.

**Figure 12 materials-14-07143-f012:**
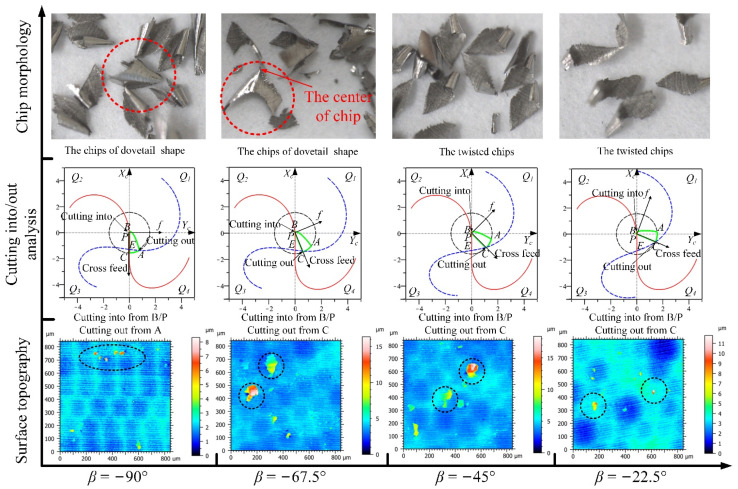
The analyses of the feed directions in the *Q*_1_ range for down-milling.

**Figure 13 materials-14-07143-f013:**
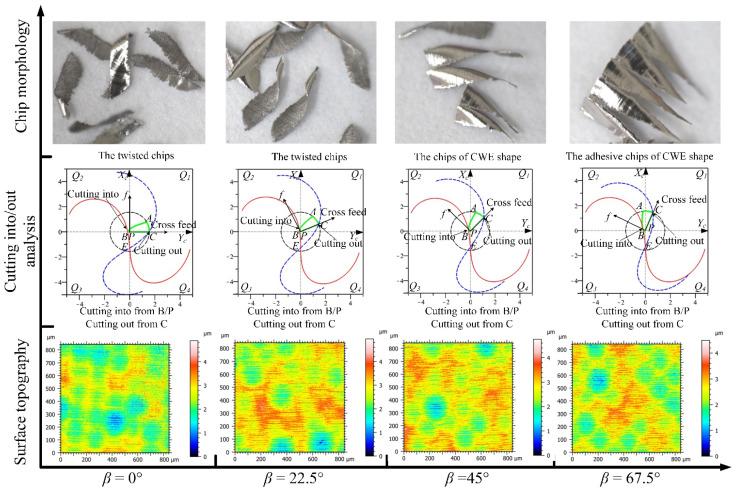
The analyses of the feed directions in the *Q*_2_ range for down-milling.

**Figure 14 materials-14-07143-f014:**
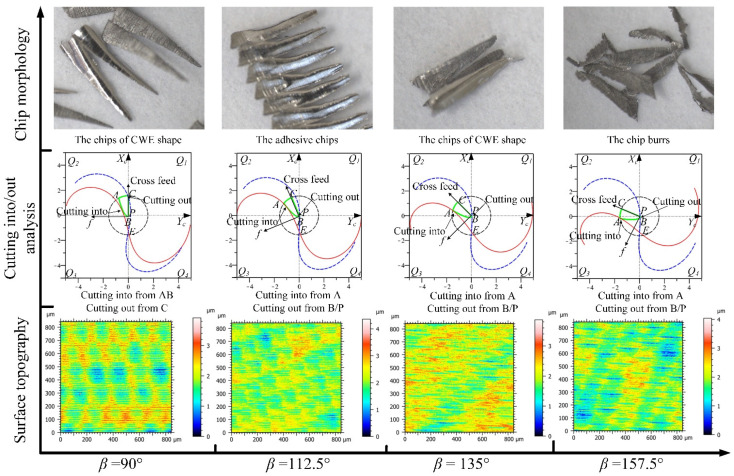
The analyses of the feed directions in the *Q*_3_ range for down-milling.

**Figure 15 materials-14-07143-f015:**
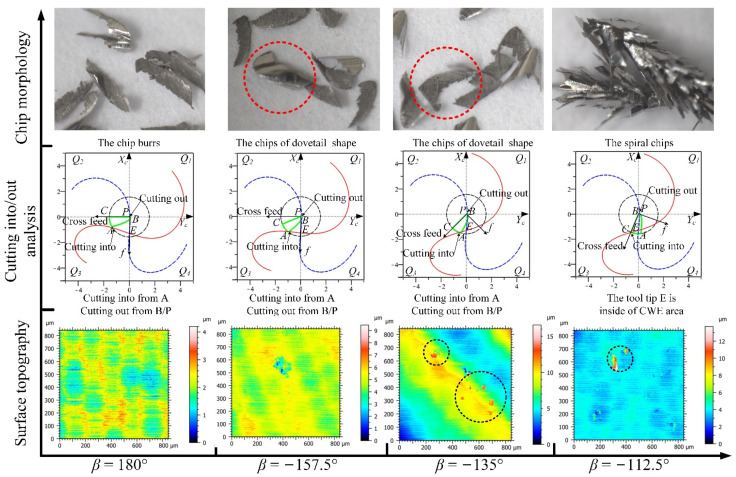
The analyses of the feed directions in the *Q*_4_ range for down-milling.

**Figure 16 materials-14-07143-f016:**
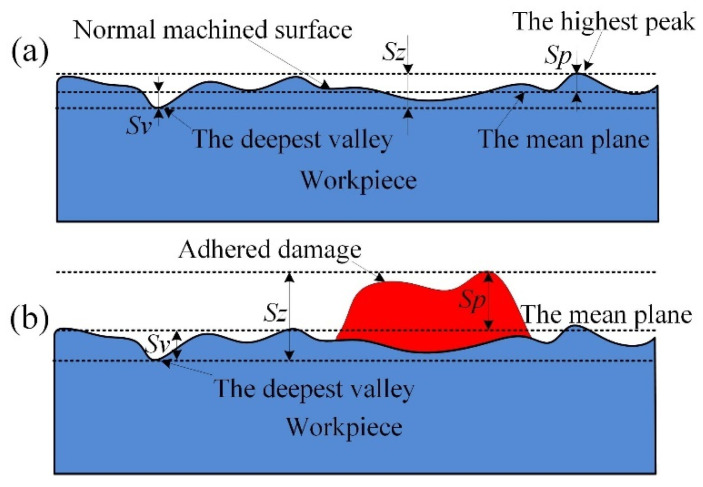
Effect of surface adhered damage on height parameters of surface topography, (**a**) normal machined surface, and (**b**) surface adhered on the machined surface.

**Figure 17 materials-14-07143-f017:**
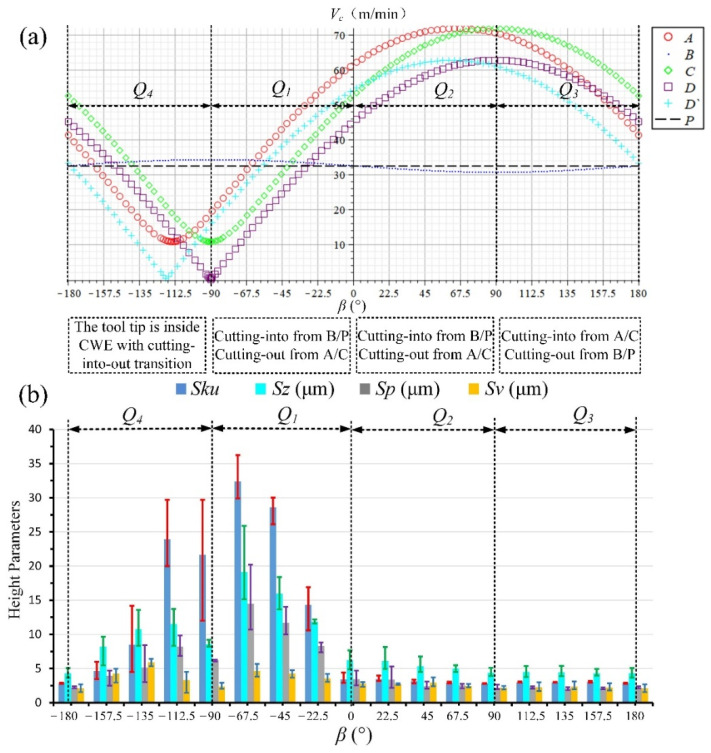
(**a**) Cutting speed variation of the critical points in the CWE area and (**b**) the selected height parameters of surface topographies.

**Figure 18 materials-14-07143-f018:**
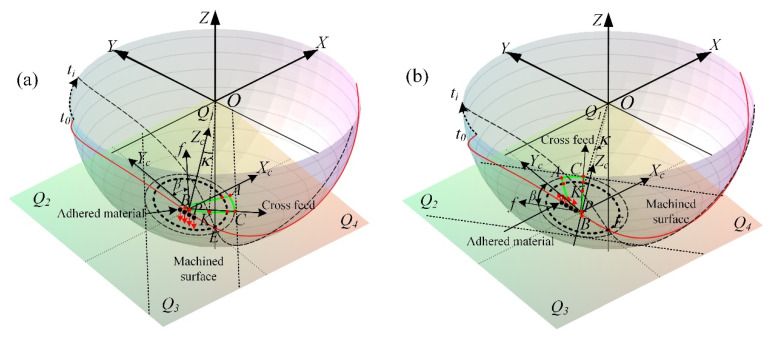
The CWE geometric analyses, (**a**) *Q*_1_ range and (**b**) *Q*_2_ range.

**Figure 19 materials-14-07143-f019:**
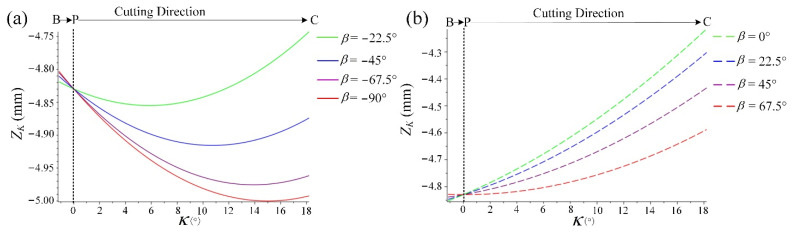
The variation trend of *Z_k_* with *κ*, (**a**) *Q*_1_ range and (**b**) *Q*_2_ range.

**Figure 20 materials-14-07143-f020:**
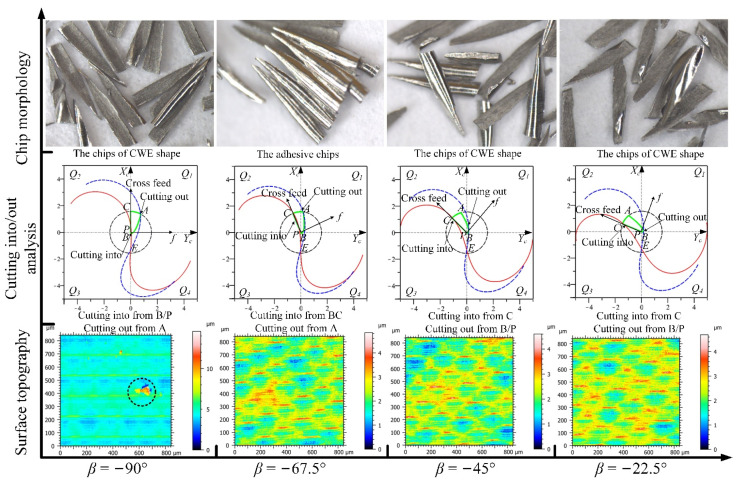
The analyses of the feed directions in the *Q*_1_ range for up-milling.

**Figure 21 materials-14-07143-f021:**
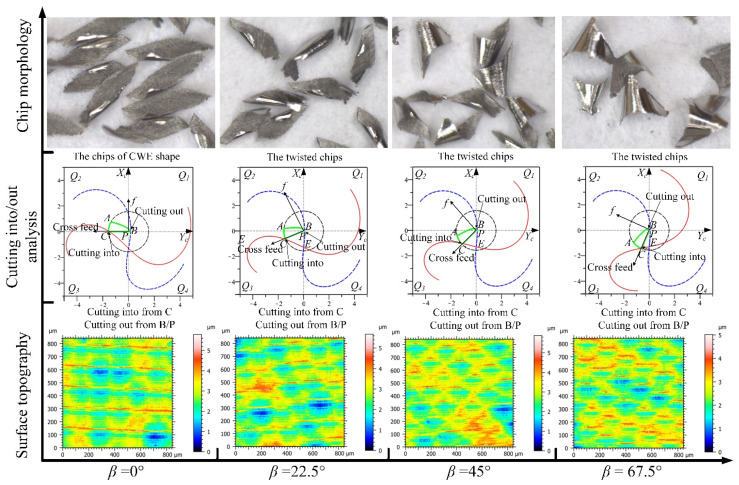
The analyses of the feed directions in the *Q*_2_ range for up-milling.

**Figure 22 materials-14-07143-f022:**
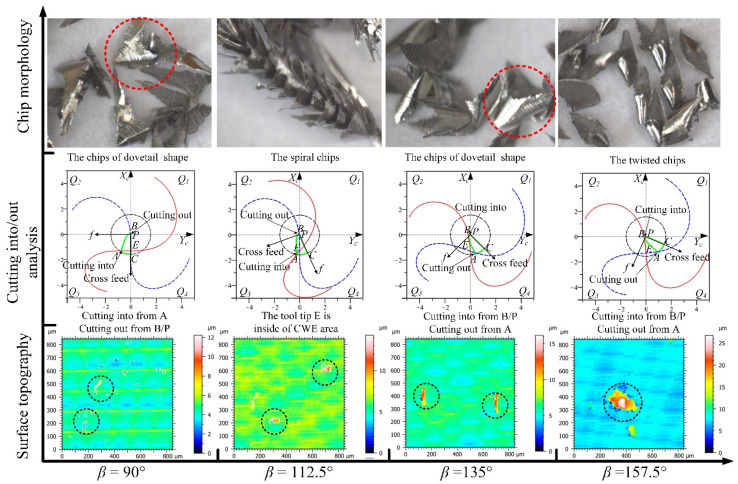
The analyses of the feed directions in the *Q*_3_ range for up-milling.

**Figure 23 materials-14-07143-f023:**
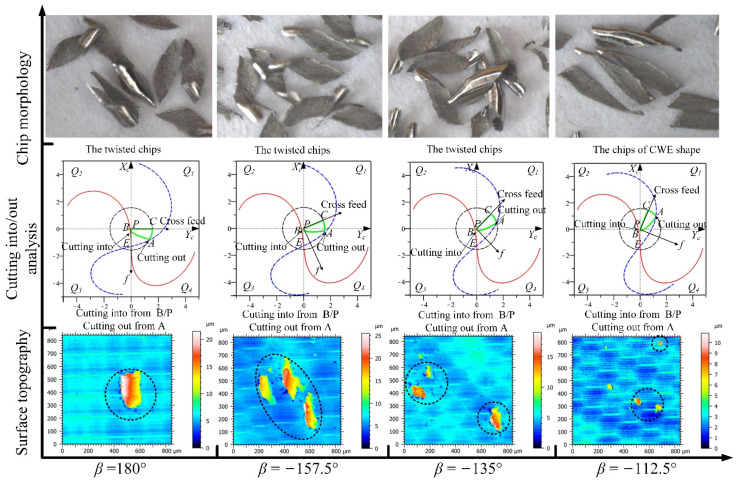
The analyses of the feed directions in the *Q*_4_ range for up-milling.

**Figure 24 materials-14-07143-f024:**
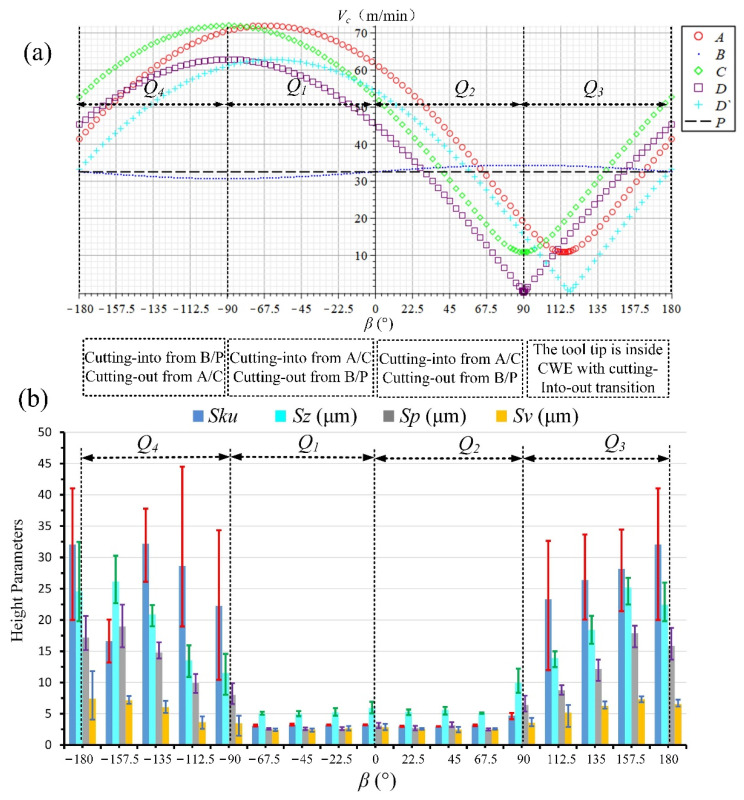
(**a**) Cutting speed variation of the critical points in CWE area and (**b**) the selected height parameters of surface topographies.

**Table 1 materials-14-07143-t001:** Chemical composition (%) of Ti–6Al–4V alloy. (Reprinted from Ref. [26].)

Element	Al	V	Fe	C	N	H	O	Ti
%	5.5~6.75	3.5~4.5	0.3	0.08	0.05	0.01	0.2	Balance

**Table 2 materials-14-07143-t002:** Mechanical and thermal properties of Ti-6Al-4V. (Reprinted from Ref. [26].)

Density (g/cm^3^)	Hardness (HB)	Modulus E (GPa)	Tensile Strength (MPa)	Thermal Conductivity (W/m.K)	Melting Point (°C)
4.42	345	113.8	995	7.3	1670

**Table 3 materials-14-07143-t003:** Specification of cemented carbide CTS20D. (Reprinted from Ref. [29].)

Parameter	Value
Tungsten carbide	90%
Cobalt	10%
Hardness (HRA)	91.9
Hardness (HV)	1600
TRS—transverse rupture strength (PSI)	580.100
Fracture toughness (MPa·m^1/2^)	10.4
Density (g/cm^3^)	14.38
Grain Size (μm)	submicron 0.8
ISO range:	K20–K40

**Table 4 materials-14-07143-t004:** Cutting conditions and case of cutting into/out for down-milling.

Cutting Conditions	Feed Ranges	Feed Directions	The Case of Cutting Into/Out
Spindle speed: *n* = 4000 rpm; Feed rate: *F* = 640 m/min; Cutting depth: *e* = 0.3 mm; Toolpath stepover: *s* = 0.15 mm; machining inclination angle: *α_p_* = 15°; Cooling mode: dry cutting;	*Q* _1_	*β* = −90°	Tool tip is inside of CWE area with cutting into/out transition
		*β* = −67.5°	Cutting into from B/P, and Cutting out from C
		*β* = −45°	Cutting into from B/P, and Cutting out from C
		*β* = −22.5°	Cutting into from B/P, and Cutting out from C
	*Q* _2_	*β* = 0°	Cutting into from B/P, and Cutting out from C
		*β* = 22.5°	Cutting into from B/P, and Cutting out from C
		*β* = 45°	Cutting into from B/P, and Cutting out from C
		*β* = 67.5°	Cutting into from B/P, and Cutting out from C
	*Q* _3_	*β* = 90°	Cutting into from AB, and Cutting out from C
		*β* = 112.5°	Cutting into from A, and Cutting out from B/P
		*β* = 135°	Cutting into from A, and Cutting out from B/P
		*β* = 157.5°	Cutting into from A, and Cutting out from B/P
	*Q* _4_	*β* = 180°	Cutting into from A, and Cutting out from B/P
		*β* = −157.5°	Cutting into from A, and Cutting out from B/P
		*β* = −135°	Cutting into from A, and Cutting out from B/P
		*β* = −112.5°	Tool tip is inside of CWE area with cutting into/out transition

**Table 5 materials-14-07143-t005:** Cutting conditions and case of cutting into/out for up-milling.

Cutting Conditions	Feed Ranges	Feed Directions	The Case of Cutting Into/Out
Spindle speed: *n* = 4000 rpm; Feed rate: *F* = 640 m/min; Cutting depth: *e* = 0.3 mm; Toolpath stepover: *s* = 0.15 mm; machining inclination angle: *α_p_* = 15°; Cooling mode: dry cutting;	*Q* _1_	*β* = −90°	Cutting into from B/P, and Cutting out from A
		*β* = −67.5°	Cutting into from BC, and Cutting out from A
		*β* = −45°	Cutting into from C, and Cutting out from B/P
		*β* = −22.5°	Cutting into from C, and Cutting out from B/P
	*Q* _2_	*β* = 0°	Cutting into from C, and Cutting out from B/P
		*β* = 22.5°	Cutting into from C, and Cutting out from B/P
		*β* = 45°	Cutting into from C, and Cutting out from B/P
		*β* = 67.5°	Cutting into from C, and Cutting out from B/P
	*Q* _3_	*β* = 90°	Tool tip is inside of CWE area with cutting into/out transition
		*β* = 112.5°	
		*β* = 135°	Cutting into from B/P, and Cutting out from A
		*β* = 157.5°	Cutting into from B/P, and Cutting out from A
	*Q* _4_	*β* = 180°	Cutting into from B/P, and Cutting out from A
		*β* = −157.5°	Cutting into from B/P, and Cutting out from A
		*β* = −135°	Cutting into from B/P, and Cutting out from A
		*β* = −112.5°	Cutting into from B/P, and Cutting out from A

**Table 6 materials-14-07143-t006:** The measured results of selected height parameters for down-milling surface topographies.

Feed Directions	*Sku*			*Sz* (μm)			*Sp* (μm)			*Sv* (μm)		
	1st	2nd	3rd	1st	2nd	3rd	1st	2nd	3rd	1st	2nd	3rd
*β* = −157.5°	4.370	5.985	3.462	4.201	3.664	5.069	2.180	2.095	2.386	2.021	1.569	2.682
*β* = −135°	6.819	4.494	14.194	9.458	9.666	5.461	4.600	4.702	2.508	4.858	4.964	2.953
*β* = −112.5°	19.985	29.699	22.076	13.584	8.317	10.398	8.407	2.993	3.998	5.969	5.325	6.400
*β* = −90°	23.160	29.699	12.016	13.712	8.314	12.510	9.841	6.840	7.978	3.871	1.474	4.532
*β* = −67.5°	36.242	29.928	31.046	8.321	9.194	8.202	6.291	6.284	6.016	2.029	2.910	2.185
*β* = −45°	30.028	26.121	29.610	16.350	15.137	25.891	12.526	10.720	20.210	3.825	4.417	5.681
*β* = −22.5°	10.574	16.915	15.456	18.394	13.665	15.849	14.040	10.007	11.107	4.354	3.658	4.742
*β* = 0°	2.932	2.894	4.421	11.795	11.591	12.183	8.744	7.388	8.806	3.051	4.203	3.377
*β* = 22.5°	3.168	3.956	3.208	4.883	6.113	7.664	2.529	3.098	4.692	2.353	3.016	2.972
*β* = 45°	3.251	2.822	3.352	4.953	8.149	5.248	2.172	5.320	2.631	2.781	2.828	2.617
*β* = 67.5°	2.822	3.061	2.858	4.760	4.498	6.756	2.070	2.103	3.077	2.691	2.395	3.679
*β* = 90°	2.764	2.838	2.861	4.499	5.493	5.009	2.103	2.760	2.760	2.396	2.733	2.249
*β* = 112.5°	3.086	2.965	2.995	3.916	5.085	4.167	1.990	2.639	2.006	1.926	2.445	2.160
*β* = 135°	2.998	3.041	2.958	4.363	3.772	5.352	2.282	2.072	2.382	2.081	1.700	2.969
*β* = 157.5°	3.036	3.179	2.942	3.873	4.040	5.384	1.833	2.101	2.272	2.040	1.939	3.112
*β* = 180°	2.899	2.929	2.773	4.039	3.964	4.943	2.243	1.952	2.099	1.796	2.012	2.844

**Table 7 materials-14-07143-t007:** Measurement results of the selected height parameters for up-milling surface topographies.

Feed Directions	*Sku*			*Sz* (μm)			*Sp* (μm)			*Sv* (μm)		
	1st	2nd	3rd	1st	2nd	3rd	1st	2nd	3rd	1st	2nd	3rd
*β* = −157.5°	13.211	16.473	20.073	25.508	30.278	22.699	18.904	22.456	15.635	6.604	7.822	7.064
*β* = −135°	26.120	37.778	32.647	18.979	21.243	22.399	13.831	14.178	16.424	5.148	7.065	5.975
*β* = −112.5°	22.462	44.487	18.960	10.897	13.805	15.946	8.325	10.025	11.370	2.572	3.780	4.576
*β* = −90°	22.022	34.336	10.429	14.570	8.031	11.969	9.878	6.547	7.685	4.692	1.484	4.284
*β* = −67.5°	3.249	3.045	2.969	4.841	5.330	4.897	2.616	2.699	2.470	2.226	2.631	2.427
*β* = −45°	3.399	3.114	3.270	4.605	5.439	4.877	2.499	2.792	2.368	2.106	2.646	2.509
*β* = −22.5°	3.268	3.140	3.193	4.693	5.111	5.891	2.371	2.410	2.865	2.321	2.701	3.026
*β* = 0°	3.270	3.294	3.169	5.796	6.912	5.054	3.085	3.537	2.687	2.710	3.375	2.367
*β* = 22.5°	3.075	3.083	2.868	5.675	5.423	4.806	3.059	2.699	2.365	2.616	2.724	2.441
*β* = 45°	3.006	2.969	2.933	5.884	6.075	4.883	2.984	3.642	2.865	2.899	2.433	2.017
*β* = 67.5°	3.030	3.273	3.165	5.021	5.195	4.987	2.558	2.658	2.303	2.463	2.537	2.683
*β* = 90°	4.788	5.134	4.128	12.225	9.316	8.356	7.900	5.965	5.362	4.325	3.350	2.994
*β* = 112.5°	25.356	32.619	11.970	14.309	14.993	12.449	8.053	8.569	9.565	6.256	6.423	2.884
*β* = 135°	33.651	20.068	25.437	16.186	18.398	20.660	10.253	12.569	13.670	5.933	5.829	6.990
*β* = 157.5°	34.446	21.405	28.679	26.726	26.398	22.470	18.942	19.069	15.650	7.785	7.330	6.820
*β* = 180°	20.018	41.030	35.036	21.489	19.818	25.987	15.188	13.663	18.749	6.300	6.155	7.238

## Data Availability

Not applicable.

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
