# Peer review of "Study on the Formation Mechanism of Surface Adhered Damage in Ball-End Milling Ti6Al4V"

_materials, 2021, doi:10.3390/ma14237143_

Round 1
Reviewer 1 Report
The reviewer comments of the paper «Study on the Formation Mechanism of Surface Adhered Damage in Ball-end Milling Ti6Al4V»- Reviewer
The authors presented an article «Study on the Formation Mechanism of Surface Adhered Damage in Ball-end Milling Ti6Al4V». In general, the article is interesting and well written and framed. However, there are several points in the article that require further explanation.
Comment 1:
The abstract needs to be improved.
Demonstrate in the abstract novelty, practical significance. Add quantitative and qualitative work results to the abstract.
Comment 2:
The introduction needs to be improved.
Now the list of references needs to be supplemented with at least 6-8 more references published over the past 5 years. Here are some recent articles:
Journal of Materials Research and Technology 2021, 11, 719–753. DOI: 10.1016/j.jmrt.2021.01.031
It is necessary to add a paragraph and a detailed analysis of the studied material of the workpiece. What difficulties are there in the machining and milling process especially? Why is this material so important? Here are just a few articles on Ti-6Al-4V:
Journal of Mechanical Science and Technology 2020, 34(9), 3767-3774. doi:10.1007/s12206-020-0828-9
Journal of Cleaner Production 2021, 281, 125374. doi:10.1016/j.jclepro.2020.125374
International Journal of Precision Engineering and Manufacturing - Green Technology 2021. doi:10.1007/s40684-021-00383-y
After analyzing the literature, show before formulating the goal of the "blank" spots. Which has not been previously done by other researchers. You must show the importance of the research being undertaken. Show what will be the new research approach in this article. You need to show a hypothesis.
Comment 3:
Sections 2 and 3 are best done as subsections of Section 2 Materials and Methods.
All abbreviations found in the text for the first time must be deciphered. For example, what is CWE? Authors should carefully address such shortcomings.
Are all figures original? If not needed appropriate citations and permissions.
It will be useful to add a table with cutting conditions and output (specify which) data.
Describe the measurement procedure in more detail. At what point in time? How is the measuring setup set up? How many repetitions of measurements? What statistical methods are used to process experimental results? Describe the experimental stand in more detail. What method of experiment planning is used and why?
Comment 4:
- Results and discussion
Add in caption cutting conditions for which each figure is obtained in the captions.
The quality and resolution figures need to be improved.
Comment 5:
It will be useful to add a section of Nomenclature in which to sign all the physical quantities and abbreviations encountered in the article. There are many physical quantities in the text and such a section will help to find the description of the necessary element.
For example,
- : Density (g/cm3)
CNC : Computer Numerical Control
etc.
Comment 6:
Conclusions.
It is necessary to more clearly show the novelty of the article and the advantages of the proposed method. Add qualitative and quantitative results of your work. What is the difference from previous work in this area? Show practical relevance.
The article is interesting, but needs to be improved. Authors should carefully study the comments and make improvements to the article step by step. After major changes can an article be considered for publication in the "Materials".
Reviewer 2 Report
The work discussed the subject of the Study on the Formation Mechanism of Surface Adhered Damage in Ball-end Milling Ti6Al4V. Presented in the manuscript research shows a good level however not in the materials science range of research. Taking care of the high level of research target to the Materials, I advise the rejection of this article.
General
The article is well written and it deserves attention, however, not in materials science, other journals seem to be more appropriate ranging the scope of research.
Reviewer 3 Report
The authors present a very interesting study on the formation mechanism of adhered surface damage in Ti6Al4V ball nose milling.
The analysis of the machining of Ti-6Al-4V parts has been widely discussed. However, the challenges of poor machined surface quality are still present in the industry.
Although it is outside the scope of this work, it is highly recommended that the authors mention in the introduction or future work the most reluctant advances in Additive Manufacturing of Ti6Al4V alloy.
- LIU, Shunyu; SHIN, Yung C. Additive manufacturing of Ti6Al4V alloy: A review. Materials & Design, 2019, vol. 164, p. 107552.
- Ghods, S., Schur, R., Schultz, E., Pahuja, R., Montelione, A., Wisdom, C., ... & Ramulu, M. (2021). Powder reuse and its contribution to porosity in additive manufacturing of Ti6Al4V. Materialia, 15, 100992.
- Bambach, M., Sizova, I., Szyndler, J., Bennett, J., Hyatt, G., Cao, J., ... & Merklein, M. (2021). On the hot deformation behavior of Ti-6Al-4V made by additive manufacturing. Journal of Materials Processing Technology, 288, 116840.
While in this document, it is determined that the formation of damage on the bonded surface is related to the adhesion of the material near the cutting edge and the cutting position within the tool rotation cycle; these problems can be overcome by the (AM).
Reviewer 4 Report
Dear author(s), please find below comments that, I hope, may justify my final evaluation of the reviewed manuscript ‘Study on the Formation Mechanism of Surface Adhered Damage in Ball-end Milling Ti6Al4V’, Manuscript ID: materials-1391461.
Generally, the paper can be classified as interesting and encouraging. However, some non-completely resolved issues make an understanding of the paper difficult and the reader confused. Please find below some suggestions, that, in my opinion, is reasonably required for an improvement of the manuscript and its further considering for publication in Materials journal:
- In the ‘Introduction’ section, especially between lines 70-71, some critical comments of the cited items might have been found. There are no slight proceedings from papers reviews and justification of the results (methods) presented in the manuscript.
- Section with abbreviations and shortcuts should be provided that, in some cases, the manuscript is difficult to follow. For example, the CWE (cutter workpiece engagement) was introduced in line 119 but was firstly mentioned in line 95.
- The keywords might have been provided with more examples. In my feelings, this should be re-worked to make more clarification at the beginning of the manuscript reading.
- There is no detail on how was the Talysurf CCI measurement provided. There is no information about measurement uncertainty and, generally, measurement errors, like noise, individual peaks (spikes), resolving the problem of unmeasured points. In some of the figures containing surface topography unmeasured points were included, in some not, it was not unified. When errors in surface topography measurement occur, results are difficult to be reliable. Moreover, the areas of measurement were not defined, e.g. centre-part, edge-part of the probe. It is my understanding that some relocation method was provided. If so, it must be mentioned. Otherwise, how can be received results classified as plausible? Furthermore, in figure 12, there are topographies with unmeasured points and surfaces without this type of point(s). Were they digitally removed or did not occur while the measurement process is provided?
- Isn't the first part (lines 447-448) of conclusion no1 based on the previous studies (from previously published papers)?
- In ‘Conclusion’ there is no word about surface topography (parameters). From that state, it seems like a surface topography measurement is not necessary. Otherwise, some remarks should be placed in this part of the manuscript considered. Furthermore, it looks like its (surface topography) measurement was not necessary. Some conclusion(s) should be indicated.
- In the text, it seems like a surface topography is analysed only to describe more damage (lines 424-425) than influence (changes) in surface topography parameters values.
- How were the height parameters selected? Why a kurtosis (Sku), peak (Spk) and valley (Sv) depth/height were measured and considered instead of all of the other (ISO 25178) parameters? For example, Std (texture direction). It was only a slight (lines 324-329) justification in the manuscript that did not clarify all of the reader doubts.
- Can be a vibration of the system taken into consideration?
- Why measured area was 800 µm x 800 µm? Analysed details were extracted from measured data? It was not introduced and justified. How was the spacing defined? No measurement details make the reader confused, especially with metrology performances.
- How was the equation (e.g. from (2) to (7)) found? Proposed by authors or received from the literature review? If received from other papers, they should be cited.
Moreover, some editorial issues can be also found, as below:
- The numbering of citations should be in the order in which they are placed, e.g. line 61 ([31]).
- Values on the axes (X, Y, Z) should be unified to indicate difference more clearly, e.g. in Figure 9.
- What do values (from -4 to 4) in the axes in the second row in Figures 12-15 (and 19-22 simultaneously) indicates? Are they coordinates or anything else?
From all of the above, the manuscript can be further considered, however, many suggested issues must be significantly improved to make the paper suitable for publication in the Materials journal.
Round 2
Reviewer 1 Report
The authors have improved the article. However, some comments received insufficient responses. Therefore, before publishing, authors need to:
1. Add the main quantitative and qualitative conclusions of the article in the abstract section. This was suggested to the authors in the first round of revision, but was not properly added.
2. It is necessary to expand all abbreviations and physical quantities in the nomenclature. For example, what is "Talysurf CCI"? Also in the nomenclature there is no Spindle speed n, and some other parameters of tables 3 and 4,5. By the way, rpm should be used for this parameter, not r / min.
Check carefully the entire text of the article, make these additions.
After these improvements, the article can be accepted for publication.
Reviewer 4 Report
Dear authors, thank you for your responses to the comments raised in the first round of the review process. All of them improve the paper, however, some of the issues must be clarified, as follows:
- In line 246 there was mentioned ‘pixels’ considering analysis of the results of surface topography measurements. How can be surface topography measurements results presented, in ‘pixels’ or ‘points’, was not clarified? Even the results are presented in the colour map, usually (excluding image-based measurement techniques), we receive an array of points (values) and, simultaneously, the image is only a colour-result of the values. It should be presented more precisely, especially that both white light interferometry (array of points/values) and super-depth microscopy (image) was applied. The reader can feel confused when a not precisely defined method is received.
- How form (line 255) was removed? It was not precisely defined. Levelling and areal form removal are two different digital actions made on the raw measured data. In fact, usually form is removed after a levelling process, nevertheless, they can be provided with different methods, e.g. levelling in commercial software is received by application of a polynomial plane of 1rst order but, respectively, a form can be eliminated by polynomials of higher (2nd or greater) degree or, often, by various procedures, e.g. digital filter, based on the Gaussian function. Therefore, at this stage of analysis, errors can be received when separating form, waviness and roughness is performed more than a levelling process. Please look for some suggestions for the following papers:
(1) https://doi.org/10.1177%2F0954405410397235
(2) https://doi.org/10.1016/j.acme.2016.12.004
- Can be the precision of the measurement equipment indicated with a consideration of the measurement uncertainty and noise? Each measurement is affected by various types of errors, generally divided into those typical for the measuring method (6), caused by the digitization process (7), errors obtained during data processing (8) and other errors. Therefore, the sentence from lines 256-258 should be verified, please look for the following papers:
(3) https://doi.org/10.1016/j.measurement.2006.07.009
(4) https://doi.org/10.3390/met11010143
- According to the sentence from lines 374-387, in fact, the severity of surface adhered damage can affect the values of an Sp and Sz parameters, but, respectively, the ‘position’ of the mean line (reference line or, simplifying for an areal surface topography measurement and analysis, reference plane) can be also influenced by this factor. Some, even short, words should be also mentioned. Moreover, when the mean plane, Sp and especially Sz parameters are modified, the value of the Sv parameter can also (even slightly) be changed. Therefore, in my feeling, the group of Sz (Sz = Sp + Sv) parameters, should be classified as those influenced by the surface adhered damage effect. It can be also mentioned, in a ‘Conclusion’ part, lines 521-525.
- The sentence ‘height parameters’ in Figure 24 (lines 512) should be modified to the ‘selected height parameters’ that the Sku and Sz are only a few from the group of height parameters.
- All of the mentioned above papers should be considered, however, it is not required to be cited in a manuscript. Reviewer leaves that for authors as a freely suggestion. Nevertheless, much valuable information may be found for authors to improve some of the issues raised in a revised review.
From all of the above issues, the revised manuscript can be further considered for publication in the Materials journal, nevertheless, comments must be responses and required action made on the manuscript before further processing.
